# An assessment of the value of deep neural networks in genetic risk prediction for surgically relevant outcomes

Mathias Aagaard Christensen[1,2], Arnór Sigurdsson[3,4], Alexander Bonde[1,2], Simon Rasmussen[4], Sisse R. Ostrowski[5,6], Mads Nielsen[7], Martin Sillesen[1,2,5]*

1 Department of Organ Surgery and Transplantation, Copenhagen University Hospital, Rigshospitalet, Copenhagen, Denmark, 2 Center for Surgical Translational and Artificial Intelligence Research (CSTAR), Copenhagen University Hospital, Rigshospitalet, Copenhagen, Denmark, 3 Novo Nordisk Foundation Center for Basic Metabolic Research, Faculty of Health and Medical Sciences, University of Copenhagen, Copenhagen, Denmark, 4 The Novo Nordisk Foundation Center for Genomic Mechanisms of Disease, Broad Institute of MIT and Harvard, Cambridge, Massachusetts, United States of America, 5 Department of Clinical Medicine, Faculty of Health and Medical Sciences, University of Copenhagen Medical School, Copenhagen, Denmark, 6 Department of Clinical Immunology, Rigshospitalet, Copenhagen University Hospital, Copenhagen, Denmark, 7 Department of Computer Science, University of Copenhagen, Copenhagen, Denmark

* Martin.Sillesen@regionh.dk

**Data Availability Statement:** Data is not publicly available but can be applied for at https://www.ukbiobank.ac.uk/enable-your-research/apply-for-access. Analytic methods will be made public at

## Abstract

### Introduction

Postoperative complications affect up to 15% of surgical patients constituting a major part of the overall disease burden in a modern healthcare system. While several surgical risk calculators have been developed, none have so far been shown to decrease the associated mortality and morbidity. Combining deep neural networks and genomics with the already established clinical predictors may hold promise for improvement.

### Methods

The UK Biobank was utilized to build linear and deep learning models for the prediction of surgery relevant outcomes. An initial GWAS for the relevant outcomes was initially conducted to select the Single Nucleotide Polymorphisms for inclusion in the models. Model performance was assessed with Receiver Operator Characteristics of the Area Under the Curve and optimum precision and recall. Feature importance was assessed with SHapley Additive exPlanations.

### Results

Models were generated for atrial fibrillation, venous thromboembolism and pneumonia as genetics only, clinical features only and a combined model. For venous thromboembolism, the ROC-AUCs were 60.1% [59.6%-60.4%], 63.4% [63.2%-63.4%] and 66.6% [66.2%-66.9%] for the linear models and 51.5% [49.4%-53.4%], 63.2% [61.2%-65.0%] and 62.6% [60.7%-64.5%] for the deep learning SNP, clinical and combined models, respectively. For atrial fibrillation, the ROC-AUCs were 60.3% [60.0%-60.4%], 78.7% [78.7%-78.7%] and

github.com at request. Requests to access these datasets should be directed to https://www.ukbiobank.ac.uk/enable-your-research/apply-for-access. The de-identified dataset used for this study can be obtained from the authors, provided written authorization from UK Biobank can be obtained. Authors are not allowed to share data without express permission from this governing body. The UK Biobank can be contacted at https://www.ukbiobank.ac.uk/learn-more-about-uk-biobank/contact-us. Enquiries from researchers about applying for access should be directed to the Access Management Team – email: access@ukbiobank.ac.uk. The data used is owned by the third-party UK Biobank Consortium and not by the authors. The data can be accessed in the same manner as the authors by the above mentioned information. The authors did not have any special access privileges, and anyone is able to apply for data access in the same way as the authors.

**Funding:** MS received a grant from the Novo Nordisk Foundation (Grant #NNF20SA0062879). https://novonordiskfonden.dk/. The funder did not play any role in the study design, data collection or analysis. The funder did not play any role in the decision to publish or in the preparation of the manuscript.

**Competing interests:** The authors have declared that no competing interests exist.

80.0% [79.9%-80.0%] for the linear models and 59.4% [58.2%-60.9%], 78.8% [77.8%-79.8%] and 79.8% [78.8%-80.9%] for the deep learning SNP, clinical and combined models, respectively. For pneumonia, the ROC-AUCs were 50.1% [49.6%-50.6%], 69.2% [69.1%-69.2%] and 68.4% [68.0%-68.5%] for the linear models and 51.0% [49.7%-52.4%], 69.7% [.5%-70.8%] and 69.7% [68.6%-70.8%] for the deep learning SNP, clinical and combined models, respectively.

## Conclusion

In this report we presented linear and deep learning predictive models for surgery relevant outcomes. Overall, predictability was similar between linear and deep learning models and inclusion of genetics seemed to improve accuracy.

## Introduction

Worldwide, more than 310 million surgeries are performed each year, addressing an estimated 11% of the global burden of disease [1, 2]. While most surgical patients proceed to an uneventful recovery, current estimates indicate that roughly 4% die as a direct or indirect result of surgery, while up to 15% experience a postoperative complication (PC), prolonging hospital length-of-stay with consequential morbidity [2].

While some Improvement in PCs following the implementation of approaches such as Enhanced Recovery after Surgery (ERAS) protocols have been well documented, the incidences of PCs have remained remarkably stable over the last decade [3]. As such, a stable subset of patients still experiences PCs, suggesting that this patient group could benefit from a deviation from the current one-size-fits all approach deployed by most ERAS protocols and a move towards a precision medicine approach in the surgical setting.

However, to achieve this goal, risk predictions models are, needed to identify which patients will fail standard ERAS protocols.

To this end, many risk assessment tools have been fielded to identify at-risk patients including the regression-based American College of Surgeons National Surgical Quality Improvement Program (ACS-NSQIP) risk calculator as well as newer machine learning approaches investigating the value of random forests or deep neural networks (DNNs) [4, 5]. However, these models are, limited by the fact that they only perform predictions on available clinical data, which only provides insight into a fraction of the driving factors that increase patients' risks of developing PCs.

As such, recent data have suggested that genetic susceptibility could, in part, be a modifier of an individual's risk of PCs, thus opening the potential for adding genetic data points to risk prediction models in order to improve model performance [6, 7].

Genetic variations are increasingly being recognized as an important modality for various surgical adverse events including venous thromboembolisms, renal complications and cardiac arrythmias [6, 8, 9]. However, it is currently not clear to what degree genetic susceptibility contributes to the overall risk compared with other well-known clinical risk factors. Furthermore, as genetic susceptibility may include complex non-linear effects such as newly identified complex interactions between genes that lie far from each other in the human genome, optimal modelling strategies remain unknown. As such, whether legacy risk prediction approaches such as the linear Polygenic Risk Scores (PGS), traditionally utilized to assess an overall genetic

risk composition and weighted sum for the phenotype in question, could be inferior to a DNN approach, is currently unknown [10].

In this study, we sought to assess whether DNNs can outperform a classic PGS approach [11]. To illustrate this assessment, we target three high impact PCs with proven genetic susceptibility, post operative pneumonia, postoperative venous thromboembolisms (pVTE), and atrial fibrillation. Furthermore, we investigate whether single nucleotide polymorphisms (SNPs) highlighted as driving the phenotype, differ between DNN and PGS approaches, thus potentially indicating that non-linear genotype-phenotype associations can be identified by the DNN approach.

We hypothesize that DNNs will achieve superior predictive performance in predicting the genotype-associated risk of these PCs compared with a linear PGS, and that the DNN models will highlight a different subset of important SNPs compared with a linear PGS.

## Methods

This study utilized genotype data from the United Kingdom biobank (UKB) consortium and adheres to the latest Polygenic Risk Score Reporting Standards [12, 13]. Access to the UKB data was approved by the consortium (Study ID #60861). Under Danish law, the study was exempt from ethical board approval due to the anonymized nature of the dataset.

We conducted a comparative study of different methodologies for genotyping risk prediction and Single Nucleotide Polymorphism (SNP)-identification in a general as well as a surgical, national cohort.

For the initial approach, we conducted standard GWAS-analyses without covariates on the chosen phenotypes with a high prevalence following surgery. Details for the GWAS are described below. These phenotypes included venous thromboembolisms (VTE), atrial fibrillation (AF) and bacterial pneumonia.

UKB has more than 500,000 individuals enrolled and consented across the United Kingdom of the age from 40 to 69. Patients were invited for participation through National Health service (NHS) registries and asked to fill surveys on basic demographic data, general lifestyle measures as well as medical history. Inclusion of all participants took place from 2006 to 2010.

### Identification of cohort

All patients with available genomic data in the UKB were initially included for analysis. Cases were identified depending on the phenotype in question. For AF, VTE and pneumonia, cases were defined using relevant *International Statistical Classification of Disease*, *9th revision* (ICD-9) and ICD-10 codes.

The phenotypes in question were identified with the ICD-9 and ICD-10 codes listed in S1 Table. The cohorts were split into training/validation and test sets. The training/validation set consisted of all non-surgical patients and a random sample of 80% of the surgical cohort. The test set consisted of the remaining 20% of the surgical cohort. Surgery was defined with the OPCS-4 codes listed in S2 Table. The post-surgical phenotypes were defined with the same ICD-codes as above registered up to 30 days after the given procedure. For AF, only first-time diagnoses were counted as post-surgery cases. For VTE and pneumonia, any diagnoses within 30 days were counted as cases, regardless of previous history.

For each outcome of interest (pAFLI, pVTE and pneumonia, both deep learning and linear models were created using three distinct input strategies (see below for model descriptions):

1. A genotype only model: using only the identified SNPs (see below) as input (SNP model)

2. A clinical data only model: using only clinical data as input (Clinical model)

3. A combined model: using both SNPs and clinical data as input (Combined model)

Input SNPs were the top 100 SNPs from the discovery GWAS for each phenotype of interest, with clinical data including demographics and comorbidities (S3 Table) and combined models including both genetic and clinical data.

The majority of the individuals included are of self-reported White-British ethnic background, with only a minority being of mixed, Asian or Black self-reported ethnicity. All self-reported backgrounds were included for analysis.

## Quality control

The first 50,000 individuals included in UKB were genotyped using the Applied Biosystems UK BiLEVE Axiom Array. The remaining were genotyped using the Applied Biosystems UK Biobank Axiom Array. The two array types are equal, and the differences are not of significance. The arrays interrogated 850,000 SNPs in total. To account for potential biases, patients with outlying heterozygosity rates, cryptic relatedness (PIHAT cut-off 0.2) and sex discrepancies in data were excluded. To ensure that only participants with high-quality genomic information were included for analysis, everyone with a genotyping rate of 98% or less were excluded. To ensure that only high-quality genetic variants were left for analyses, a missingness rate of 2% were used as a cut-off point. Lastly, a Minor Allele Frequency (MAF) of > 5% was used, and variants found not to be in Hardy-Weinberg equilibrium were excluded (threshold: $1 \times 10^{-6}$ for both cases and controls).

## GWAS

The initial GWAS-analyses were analyzed using a mixed linear model (MLM) approach. GCTA version 1.93 beta for Windows was used to conduct the analyses. The MLM-model was created using fastGWA with a sparse genetic relationship matrix (GRM) with non-imputed data from the UKB. For all the phenotypes analyzed in the respective GWAS, the 100 most significant SNPs were included in the genetic and mixed models. The choice to utilize only the top 100 SNPs was made to optimize the balance between predictive power and keeping the model computational pragmatic. SNPs are referenced using the dbSNP (rs) reference number. The cohorts were split into training/validation and test sets before, and only the training data was used for the initial GWAS-models. Relevant GWAS plots, including Manhattan and Quantile-Quantile (QQ) plots were generated using qqman (R version 4.0.2) [14]. Performance plots including the Receiver Operator Characteristics of the Area Under the Curve (ROC-AUCs), Area Under the Precision-Recall Curve (PRAUC) and heatmaps were created using Scikit-learn 1.2.1 (Python 3) [15].

## Linear Polygenic risk score (PGS) modelling approach

A linear PGS was generated using the logistic regression module as implemented in scikit-learn 1.2.1 for Python 3. Models were created with both L1 (lasso) and L2 (ridge) regularization. Feature importance was determined by coefficients of the SNPs.

## Deep neural network (DNN) modelling approach

All DNN models were implemented using EIR (version 0.1.25-alpha) [16]. EIR is a framework that incorporates genetic, clinical, image, sequencing, and binary data for supervised training of deep learning models. A held-out test set was used for all models to obtain a final performance after training and validation. The Cross Entropy loss was employed during training for the classification tasks. All models were trained with a batch size of 64. During training, plateau

learning rate scheduling was used to reduce the learning rate by a factor of 0.2 if the validation performance had not improved for 10 steps, with a validation interval of 500 steps. Early stopping was used to terminate training when performance had not improved with a patience of 16 steps. The early stopping criterion was activated after a buffer of 2,000 iterations. All models were trained with the Adam optimizer with a weight decay of $1 \times 10^{-4}$ and a base learning rate of $1 \times 10^{-3}$ [17]. For the neural network models, we augmented the genotype input by randomly setting 40% of the SNPs as missing in the one-hot encoded array. All DNN models utilize the genome-localnet (GLN) architecture for the genotype feature extraction [16]. The same cohort splits were used as in the linear PGS-approach. Importance of features were determined using SHapley Additive exPlanations (SHAP) values [18].

## Results

### Cohort

We identified 488,377 patients in the UKB with available genetic and relevant phenotypic data, with 446,180 patients available for analyses after genetic quality measures were applied and were used for both the linear and deep learning modelling approaches.

For the outcomes of interest, 19,704 had a diagnosis of AF, 9,101 had a diagnosis of VTE and 13,757 had a diagnosis of pneumonia overall in the UKB. The selection of the cohort and SNPs is depicted in Fig 1.

### Linear models

**Atrial fibrillation.** Baseline characteristics are listed in Table 1. The SNP model reached a ROC-AUC of 60.3 [95% CI, 60.0%-60.4%]. All individuals were classified as not having AF. The PRAUC was 0.09. The clinical model reached a ROC-AUC of 78.7% [95% CI, 78.7%-78.7%] with a recall of 9% and a precision of 53%. The PRAUC was 0.25. The combined model reached a ROC-AUC of 80.0% [95% CI, 79.9%-80.0%] with a recall of 9% and a precision of 56%. The PRAUC was 0.28. All performances are depicted in Fig 2A. The SNPs and the associated genes with the highest feature importance are listed in Table 2.

**Venous thromboembolism.** Baseline characteristics for VTE are listed in Table 3. The SNP model reached a ROC-AUC of 60.1% [95% CI, 59.6%-60.4%]. All individuals were classified as not having VTE. The PRAUC was 0.04. The clinical model reached a ROC-AUC of 63.4% [95% CI, 63.2%-63.4%]. All individuals were classified as not having VTE. The PRAUC was 0.04. The combined model reached a ROC-AUC of 66.6% [95% CI, 66.2%-66.9%]. All individuals were classified as not having VTE. The PRAUC was 0.05. All performances are depicted in Fig 2B. The SNPs and the associated genes with the highest feature importance are listed in Table 4.

**Pneumonia.** Baseline characteristics are listed in Table 5. The SNP model reached a ROC-AUC of 50.1% [95% CI, 49.6%-50.6%]. All individuals were classified as not having pneumonia. The PRAUC was 0.04. The clinical model reached a ROC-AUC of 69.2% [95% CI, 69.1%-69.2%]. All individuals were classified as not having pneumonia. The PRAUC was 0.12. The combined model reached a ROC-AUC of 68.4% [95% CI, 68.0%-68.5%] with a recall of 0.01 and a precision of 0.5. The PRAUC was 0.11. The SNPs and the associated genes with the highest feature importance are listed in Table 6. All performances are depicted in Fig 2C.

### Deep learning models

**Atrial fibrillation.** The SNP model reached a ROC-AUC of 59.4% [95% CI, 58.2%-60.9%] in the test set. Recall was 42.8% and precision was 8.7%, with the area under the precision-

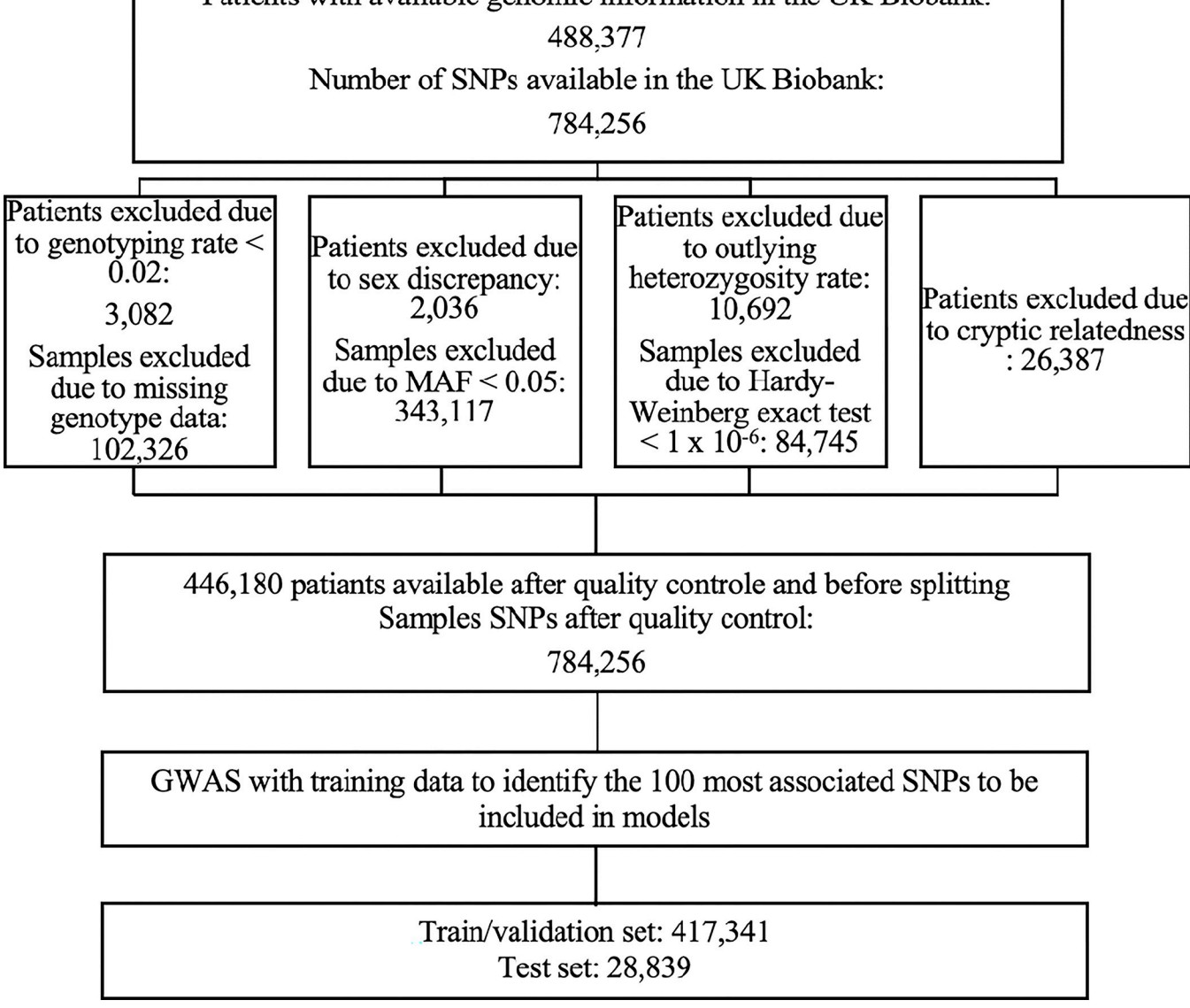

**Fig 1. Selection and quality control steps of individuals and SNPs in the UKB.**

**Table 1. Baseline characteristics for atrial fibrillation cohort.**

| Characteristic | AF | No AF |
|---|---|---|
| | N = 19,900 | N = 426,280 |
| Age, mean, SD | 62.2 ± 5.9 | 56.4 ± 8.0 |
| Female, N, % | 6,623 (33.2) | 233,351 (54.7) |
| BMI, mean, SD | 27.4 ± 5.4 | 29.1 ± 1 |
| Previous or current smoker, N, % | 9,633 (48.4) | 159,702 (37.5) |
| Previous or current cancer, N, % | 2,307 (11.6) | 35,338 (8.3) |
| Heart failure, N, % | 2,922 (14.7) | 3,252 (0.7) |
| Hypertension, N, % | 12,385 (62.2) | 86,999 (20.4) |

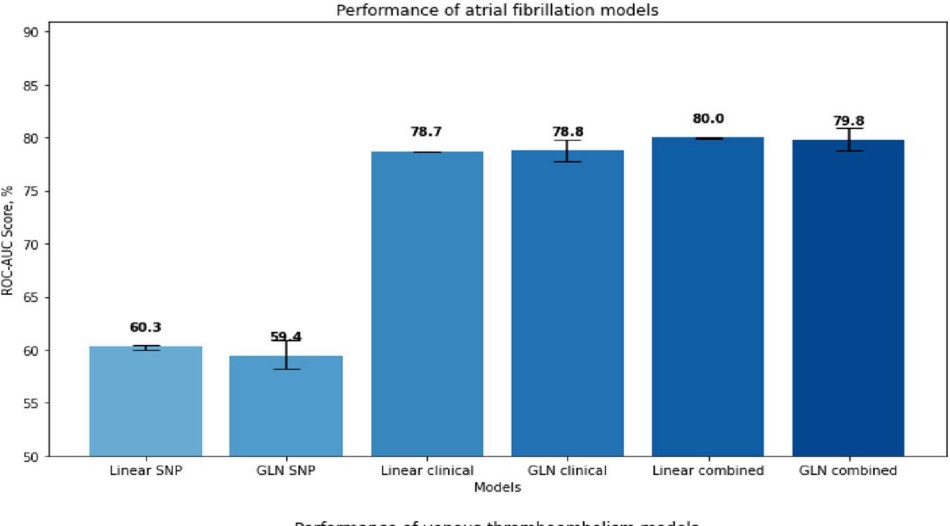

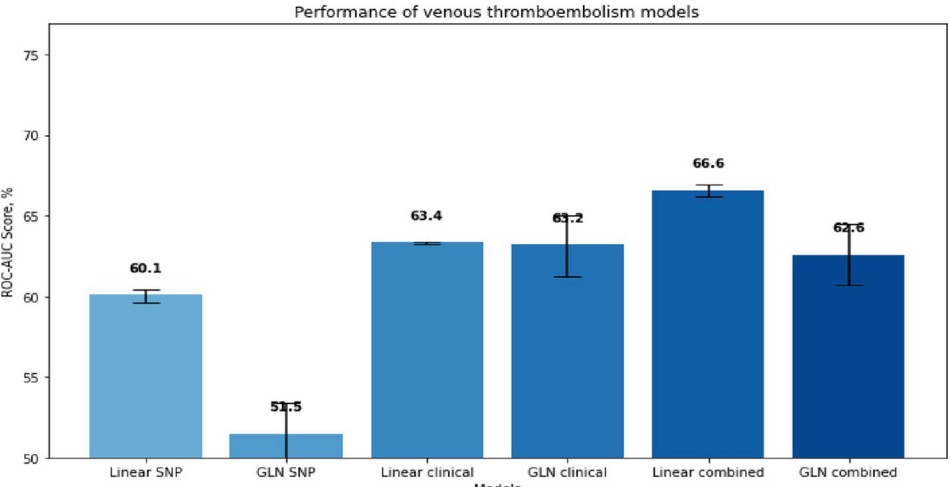

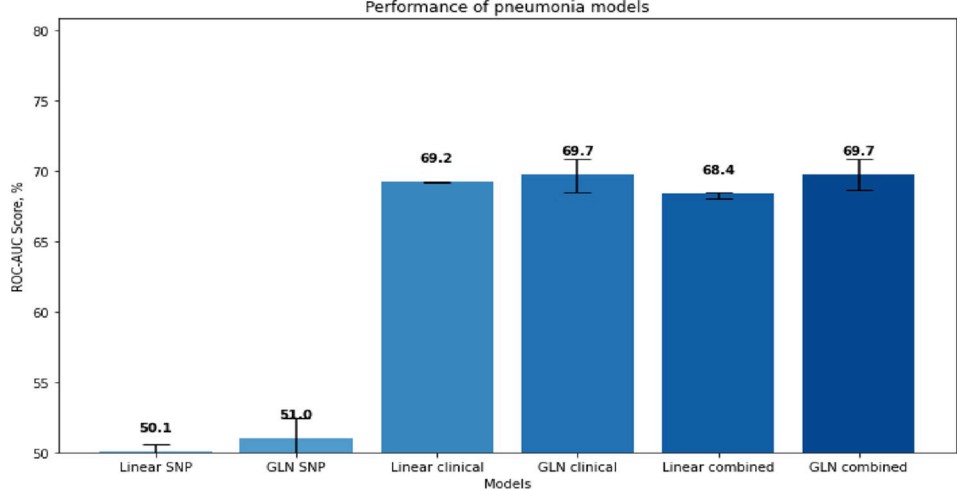

**Fig 2. A:** Bar plot of ROC-AUCs of all atrial fibrillation models. **B:** Bar plot of ROC-AUCs of all venous thromboembolism models. **C:** Bar plot of ROC-AUCs of all pneumonia models.

**Table 2. Table of the SNPs with the highest feature importance for the genetic, and mixed atrial fibrillation linear models.**

| Genetic, SNP | Coefficient (positive prediction) | Gene | Mixed, SNP | Coefficient (positive prediction) | Gene |
|---|---|---|---|---|---|
| rs17042171 | 0.282965 | Intergenic | rs17042171 | 0.341726 | Intergenic |
| rs10033464 | 0.218606 | Intergenic | rs10033464 | 0.248286 | Intergenic |
| rs3731748 | 0.182317 | *TTN/TTN-AS1* | rs3731748 | 0.232892 | *TTN/TTN-AS1* |
| rs2723065 | 0.160737 | *LINC02576* | rs3829747 | 0.205116 | *TTN/TTN-AS1* |
| rs13390491 | 0.157113 | *TTN* | rs13390491 | 0.176328 | *TTN* |

recall curve (PRAUC) being 0.09. The clinical model reached a ROC-AUC of 78.8% [95% CI, 77.8%-79.8%] with recall and precision of 72.0% and 13.5%, respectively, with the PRAUC being 0.25. The combined model reached a ROC-AUC of 79.8 [95% CI, 78.8%-80.9%] with a recall and precision of 74.8% and 13.5%, respectively, with the PRAUC being 0.27. The SNPs and the associated genes with the highest feature importance are listed in Table 7. All performances are depicted in Fig 2A.

**Venous thromboembolism.** The SNP model reached a ROC-AUC of 51.5% [95% CI, 49.4%-53.4%] with a recall of 50.8% and precision of 4% and a PRAUC of 0.03. The clinical model reached a ROC-AUC of 63.2% [95% CI, 61.2%-65.0%] with a recall and precision of 67.5% and 4.0%, respectively, and a PRAUC of 0.05. The combined model reached a ROC-AUC of 62.6% [95% CI, 60.7%-64.5%] with a recall and precision 68.8% and 4.0%, respectively, and a PRROC of 0.05. The SNPs and the associated genes with the highest feature importance are listed in Table 8. All performances are depicted in Fig 2B.

**Pneumonia.** The SNP model reached a ROC-AUC of 51.0% [95% CI, 49.7%-52.4%] with a recall of 55.0% and precision of 5% and a PRAUC of 0.05. The clinical model reached a ROC-AUC of 69.7% [95% CI, 68.5%-70.8%] with a recall and precision of 67.7% and 7.4%, respectively and a PRAUC of 0.13. The combined model reached a ROC-AUC of 69.7% [95% CI, 68.6%-70.8%] with a recall and precision of 70.1% and 7.3%, respectively, and a PRAUC of

**Table 3. Baseline characteristics for venous thromboembolism cohort.**

| Characteristic | VTE | No VTE |
|---|---|---|
| | N = 9,193 | N = 436,987 |
| Age, mean, SD | 59.7 ± 7.1 | 56.6 ± 8.0 |
| Female, N, % | 4,156 (45.2) | 235,818 (54.0) |
| BMI, mean, SD | 29.4 ± 5.5 | 27.4 ± 4.8 |
| Previous or current smoker, N, % | 3,842 (41.8) | 165,493 (37.9) |
| Previous or current cancer, N, % | 1,403 (15.3) | 36,242 (8.3) |
| Heart failure, N, % | 482 (5.2) | 5.692 (1.3) |
| Hypertension, N, % | 3,906 (42.5) | 95,478 (21.8) |

**Table 4. Table of the SNPs with the highest feature importance for the genetic, and mixed venous thromboembolism linear models.**

| Genetic, SNP | Coefficient (positive prediction) | Gene | Mixed, SNP | Coefficient (positive prediction) | Gene |
|---|---|---|---|---|---|
| rs4524 | 0.329348 | *F5* | rs4524 | 0.304659 | *F5* |
| rs59262400 | 0.286520 | *TSPAN15* | rs59262400 | 0.300188 | *TSPAN15* |
| rs6030 | 0.240644 | *F5* | rs6050 | 0.217009 | *FGA* |
| rs6050 | 0.234128 | *FGA* | rs6030 | 0.208518 | *F5* |
| rs75112989 | 0.203980 | *ATP1B1* | rs75112989 | 0.196367 | *ATP1B1* |

**Table 5. Baseline characteristics for pneumonia.**

| Characteristic | Pneumonia | No pneumonia |
|---|---|---|
| | N = 14,101 | N = 432,079 |
| Age, mean, SD | 60.3 ± 7.2 | 56.5 ± 8.0 |
| Female, N, % | 6,101 (43.3) | 233,873 (54.1) |
| BMI, mean, SD | 28.4 ± 5.6 | 27.5 ± 4.8 |
| Previous or current smoker, N, % | 6,464 (45.8) | 162,871 (37.7) |
| Previous or current cancer, N, % | 2,144 (15.2) | 35,501(8.2) |
| Heart failure, N, % | 1,506 (10.7) | 4,668 (1.11) |
| Hypertension, N, % | 7,120 (50.5) | 92,264 (21.4) |

0.12. The SNPs and the associated genes with the highest feature importance are listed in Table 9. All performances are depicted in Fig 2C.

## Discussion

In this study, we assessed the performance of linear and deep learning models including genotypic information on specific phenotypes relevant for pAFLI, pVTE and postoperative pneumonia. Overall, we found that adding SNP data to clinical risk prediction models enhanced the predictive power.

### Modelling approaches

All three SNP linear models failed to make any meaningful hard predictions, as they classified all individuals in the cohorts as not having the disease in question. However, the separations were roughly similar to the GLN-models, as demonstrated by similarity in ROC-AUC performance, and the lack of positive predictions may be due to imbalanced data and skewed threshold for hard predictions. The GLN-models were able to classify positives correctly with just genomic information, however, the PRAUC was generally comparable between the deep learning and linear models with identical data. Given that the linear and GLN models utilize distinct

**Table 6. Table of the SNPs with the highest feature importance for the genetic, and mixed pneumonia linear models.**

| Genetic, SNP | Coefficient (positive prediction) | Gene | Mixed, SNP | | Gene |
|---|---|---|---|---|---|
| rs8062405 | 0.352614 | *ATXN2L* | rs10519203 | 0.353799 | *HYKK* |
| rs10519203 | 0.253258 | *HYKK* | rs4788102 | 0.311939 | *SH2B1* |
| rs7498665 | 0.190802 | *SH2B1* | rs72793809 | 0.187079 | *ATXN2L* |
| rs72793809 | 0.143266 | *ATXN2L* | rs7498665 | 0.152760 | *SH2B1* |
| rs8034191 | 0.133021 | *HYKK* | rs10747050 | 0.119137 | *GRIN1* |

**Table 7. Table of the SNPs with the highest feature importance for the single nucleotide polymorphism (SNP), and combined atrial fibrillation GLN-models.**

| Top SNP in SNP model | Coefficient (positive prediction) | Gene | Top SNP in combined model | Coefficient (positive prediction) | Gene |
|---|---|---|---|---|---|
| rs17042171 | 0.001151 | Intergenic | rs9940321 | 0.0006 | *ZFHX3* |
| rs52511 | 0.000464 | Not in NCBI | rs883079 | 0.000575 | *TBX5* |
| rs13376333 | 0.000352 | *KCNN3* | rs4399218 | 0.000567 | *LINC01681* |
| rs6658392 | 0.000280 | *KCNN3* | rs1895596 | 0.000548 | *TBX5* |
| rs2106261 | 0.000245 | *ZFHX3* | rs2106261 | 0.000503 | *ZFHX3* |

**Table 8. Table of the SNPs with the highest feature importance for the single nucleotide polymorphism (SNP), and combined venous thromboembolism GLN-models.**

| Top SNP in SNP model | Coefficient (positive prediction) | Gene | Top SNP in combined model | Coefficient (positive prediction) | Gene |
|---|---|---|---|---|---|
| rs8176745 | *0.000873* | *ABO* | rs505922 | *0.000388* | *ABO* |
| rs8176719 | *0.000679* | *ABO* | rs6050 | *0.000241* | *FGA* |
| rs10800454 | *0.000382* | *F5* | rs8176719 | *0.000234* | *ABO* |
| rs8106664 | *0.000381* | *SLC44A2* | rs8176745 | *0.000161* | *ABO* |
| rs10094510 | *0.000244* | *ZFPM2* | rs1593 | *0.000160* | *F11* |

tuning parameters for hard predictions, a direct comparison of recall and precision may not be critically significant, although the deep learning models generally performed better. However, as hard predictions are necessary in a clinical setting, a discussion is still warranted.

It is exemplified for AF, where the genetic linear model had a ROC-AUC of 60.3% [95% CI, 60.0%-60.4%] while the GLN had a ROC-AUC of 59.4% [95% CI, 58.2%-60.9%]. The recall, however, was 0% and 42.8%, respectively. As the linear model performed very poorly in terms of recall, this is likely to be the result of the imbalanced data and failure to capture feature relevance and possibly non-linearity and would need optimization before any form of utilization for positive prediction in a clinical setting. On the contrary, the GLN-model had a recall of 42.8% and therefore identifies more than one third of cases correctly, which heightens the likelihood of clinical meaningful utilization considering only SNPs were included in the model. However, the precision was calculated low at 8.7%, which would lead to overdiagnosis and possibly overtreatment. If used in clinical practice, it is therefore of paramount importance that any possible intervention would carry little to no risk of harm. It, however, cannot be ruled out that the differences between the models are not due to an inherent predictive advantage in the GLN-model, but simply due to different hyperparameter tuning, as indicated by the similar performance in PRAUC.

When combining SNP and clinical data for all phenotypes in question, we observed a trend towards better performance compared with SNP or clinical data only models, although most confidence intervals were overlapping with clinical model performances, with the exception being the linear combined VTE model, which performed significantly better than the GLN-model and the linear clinical model. This indicates that limited performance gains could be obtained by combining genetic and clinical data and may suggest that genotype effects may already in part be captured by diagnoses codes. Alternatively, the lack of performance improvement could be affected by limited study power due to factors such as lack of correct PC diagnoses codes, a problem often encountered when administrative codes are used for PC curation [19].

## Identified single nucleotide polymorphisms

In the GLN model, rs17042171 was the most activated SNP in regards of classifying individuals with AF. It is an intergenic variant downstream from *PITX2* which codes for the paired-like

**Table 9. Table of the SNPs with the highest feature importance for the single nucleotide polymorphism (SNP), and combined pneumonia GLN-models.**

| Top SNP in SNP model | Coefficient (positive prediction) | Gene | Top SNP in combined model | Coefficient (positive prediction) | Gene |
|---|---|---|---|---|---|
| rs17851582 | 0.000663 | *GAMT* | rs9353801 | 0.000614 | Intergenic |
| rs2505594 | 0.000549 | Intergenic | rs12459346 | 0.000579 | Intergenic |
| rs10105477 | 0.000344 | Intergenic | rs7256201 | 0.000452 | *LOC107985327* |
| rs72793809 | 0.000309 | *ATXN2L* | rs72976957 | 0.000328 | *PIAS4* |
| rs4426179 | 0.0000243 | Intergenic | rs10503997 | 0.000324 | Intergenic |

homeodomain transcription factor 2 [20, 21]. The risk allele is moderately prevalent in the European population with a prevalence of 15%. suggesting a negative selection pressure of the risk allele [22]. The most highly activated SNP in the GLN-model with importance for classifying patients for not having AF was rs17042081, a variant near 4q25, which has been extensively associated with AF in a variety of populations and is also in close proximity to *PITX2* [21, 23]. The variants most highly associated with AF in the linear model was also rs17042171. As the testing set consists of purely surgical patients, it is not unexpected that variants near 4q25 are important for the models, as the same region was the only one associated with postoperative AF in a recent GWAS-analysis from our group [7].

Besides the top variant, rs17042171, the difference in which variants show importance for the GLN and linear model, respectively, and that the GLN-models in general performed significantly better in recall compared with the matching linear, shows that non-linear interactions between genes which are potentially of great importance in the risk of a particular trait. Other explanations include non-linear effects in non-genetic features, such as age and sex, or dominant/recessive effects of the SNPs in question.

When exploring pathways and interactions for the most highly activated genes in online repositories such as the Reactome Pathway Database and BioGRID, it appears that none of the genes have previously been described to be in a direct pathway or in any kind of interaction.

The SNPs with highest importance for classifying VTE in the GLN-model was rs8176745, a synonymous variant in, in *ABO* [24]. The variant has, to our knowledge, not previously been associated with VTE [6]. The ABO blood group antigen genes are, however, amongst the most heavily associated with VTE, and it has a biological plausible explanation, thus it is expected that specific variants within these genes would play a significant role in predictive models for VTE risk.

The SNP that had the highest feature importance in the GLN-model for classifying bacterial pneumonia was rs17851582 for the SNP-model and rs9353801 for the combined model. However, as the models performed poorly in terms of the overall accuracy, there is a high likelihood that the highest activated variants are not due to genuine, biological phenomena and interactions, but rather due to chance alone. rs17851582 is a missense variant in *GAMT* which codes for the liver enzyme guanidinoacetate methyltransferase. Deficiency of the enzyme can affect brain and muscle development and lead to severe neurological problems including epilepsy [25]. However, the specific variant in our analysis is deemed to be benign [26]. rs9353801 was most important in the combined model. It has no reported clinical significance in the literature and does not lie close to any biological meaningful genes. One downstream gene, *CASC6*, codes for cancer susceptibility 6, which, although the name may indicate otherwise, has an unknown function. Although a history of cancer was included as a covariate in the initial GWAS and in both models, it cannot be ruled out that the phenotype and models are confounded by occult lung malignancy.

In the linear model, rs10519203 was the most associated with classifying pneumonia in the combined model and second most associated in the genetic model. It is an intron variant in *HYKK* and has previously been associated with lung cancer and smoking behavior, which may indicate the basis for its ability to classify pneumonia, and not an inherent increased risk to infection [27, 28].

We again observed a discrepancy between the variants with highest feature importance between the models, which suggests that complex non-linear effects may exist between the genes in question activated by the GLN-model. It should be noted that while all the phenotypes of interest in this study are complex diseases, we find it likely that the susceptibility to bacterial pneumonia is less driven by genetics compared with AF and VTE. Although certain genetic variants have been associated pneumonia susceptibility, the genetic landscape has not been

explored to the same extent as with AF and especially VTE. Consequently, the ratio of importance of the input variants compared with the clinical factors included is likely lower compared with AF and VTE, and we anticipate that future research will highlight the importance of genetics compared with clinical factors in predictive modelling.

## Potentials for clinical use

As these analyses were specifically made on phenotypes relevant for the postoperative course of surgical patients, it is of utmost importance that the models in question can be validated and potentially optimized in a specific surgical cohort. Above all, this will ascertain the utilization on this specific population, and it will further establish a foundation for the investigation into if the models are able to improve the outcomes for the phenotypes in question or be of prophylactic benefit. as multiple surgical risk predictors built on clinical data already exist, investigation of whether adding genetic data will enhance the predictability of such models could offer a promising pathway for optimizing the model performances further. It is key to establish methods to improve prediction, as the current standard of models fail to demonstrate any clear clinical benefit compared with standard practice [29]. Although multiple factors account for the current limited applicability, including lack of external validity and variance in the retro- and prospective data, a lack of important factors such as genetics may also be of significance. Further, as we present a model where a deep learning framework specifically made to incorporate genetics with clinical variables that performs better compared with a linear PGS in terms of recall and precision, but not ROC-AUC and PRAUC, it is important to consider the quality of the used software as well as the pragmatic applicability of the models in question in a clinical scenario. At this time, neither model have applicability if incorporating only the top 100 SNPs, as determined by the poor accuracy performance.

## Limitations

This study has limitations. First, all phenotypes were established using only ICD-codes which may have a low accuracy for the phenotypes in question. We suspect especially bacterial pneumonia to have an overall low accuracy due to the high hospital incidence and difference in presentation as well differences in the microbiological organism and treatments. This generates a heterogenous group which lowers the predictability and clinical utility.

Further, we assumed that the one hundred most significant SNPs from an initial GWAS for the phenotype in question would be of interest, although this was an arbitrary choosing due to the need to find an optimum between predictive power and computational efficacy. Other SNPs may also be of importance, and using a different set or potentially the entire genome has the potential to achieve similar or even better genetic predictability, although the latter would be too computational costly and of less clinical utility. A significant challenge in our study is the imbalanced data, which especially proved problematic in the linear models which all had a recall of 0. Larger cohorts, or a more balanced dataset may improve this.

## Conclusion

In conclusion, we present predictive models on surgically relevant phenotypes incorporating a small sample of genetic variants. Overall, GLN-based models performed equally when compared with linear models based on the AUC and PR metrics, with the exception being the linear combined VTE model, which performed better than the identical GLN-model and linear clinical model. Different SNPs were important for the same phenotypes between models suggesting importance off non-linear interactions. Lastly, in a comparison between clinical models with and without inclusion of SNPs, the inclusion of genetic data seemed to increase the

accuracy, especially in the linear VTE model. This is a preliminary report assessing the utility of using a small sample of SNPs for clinical risk prediction. Future research is needed to validate models in surgical cohorts and assess the utility of incorporating genetics and clinical variables in predictive models to improve surgical outcomes.

## Supporting information

**S1 Table. List of ICD-9 and ICD-10 codes used for the phenotypes in question.**
(DOCX)

**S2 Table. List of OPCS-4 codes used to define surgery.**
(DOCX)

**S3 Table. Demographics and comorbidities with ICD-codes.**
(DOCX)

## Author Contributions

**Conceptualization:** Mathias Aagaard Christensen, Arnór Sigurdsson, Alexander Bonde, Simon Rasmussen, Sisse R. Ostrowski, Mads Nielsen, Martin Sillesen.

**Data curation:** Mathias Aagaard Christensen, Martin Sillesen.

**Formal analysis:** Mathias Aagaard Christensen, Arnór Sigurdsson, Alexander Bonde, Mads Nielsen, Martin Sillesen.

**Funding acquisition:** Martin Sillesen.

**Investigation:** Mathias Aagaard Christensen, Arnór Sigurdsson, Alexander Bonde, Simon Rasmussen, Sisse R. Ostrowski, Mads Nielsen, Martin Sillesen.

**Methodology:** Mathias Aagaard Christensen, Arnór Sigurdsson, Alexander Bonde, Sisse R. Ostrowski.

**Software:** Arnór Sigurdsson, Simon Rasmussen, Mads Nielsen.

**Supervision:** Arnór Sigurdsson, Alexander Bonde, Simon Rasmussen, Sisse R. Ostrowski, Mads Nielsen, Martin Sillesen.

**Visualization:** Mathias Aagaard Christensen.

**Writing – original draft:** Mathias Aagaard Christensen.

**Writing – review & editing:** Arnór Sigurdsson, Alexander Bonde, Simon Rasmussen, Sisse R. Ostrowski, Mads Nielsen, Martin Sillesen.

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
