## [Decision Letter · Decision Letter 0]

29 Feb 2024

PONE-D-23-35139An assessment of the value of deep neural networks in genetic risk prediction for surgically relevant outcomesPLOS ONE

Dear Dr. Sillesen,

Thank you for submitting your manuscript to PLOS ONE. After careful consideration, we feel that it has merit but does not fully meet PLOS ONE’s publication criteria as it currently stands. Therefore, we invite you to submit a revised version of the manuscript that addresses the points raised during the review process.

We look forward to receiving your revised manuscript.

Kind regards,

Xiang Zhu

Academic Editor

PLOS ONE

Journal Requirements:

3. In the online submission form, you indicated that data is not publicly available but can be applied for at https://www.ukbiobank.ac.uk/enable-your-research/apply-for-access. Analytic methods will be made public at github.com at request. Requests to access these datasets should be directed to https://www.ukbiobank.ac.uk/enable-your-research/apply-for-access.

4. Please amend your manuscript to include your abstract after the title page.

5. We notice that your supplementary [figures/tables] are included in the manuscript file. Please remove them and upload them with the file type 'Supporting Information'. Please ensure that each Supporting Information file has a legend listed in the manuscript after the references list.

Additional Editor Comments :

In addition to address the critical comments from both Reviewers in full, I would encourage that the authors follow the latest version of the Polygenic Risk Score Reporting Standards (PMID: 33692554) to ensure the methodological transparency and scientific rigor in the revised manuscript.

Reviewers' comments:

Reviewer's Responses to Questions

**Comments to the Author**

1. Is the manuscript technically sound, and do the data support the conclusions?

Reviewer #1: Yes

Reviewer #2: Partly

2. Has the statistical analysis been performed appropriately and rigorously? 

Reviewer #1: Yes

Reviewer #2: No

3. Have the authors made all data underlying the findings in their manuscript fully available?

Reviewer #1: Yes

Reviewer #2: No

4. Is the manuscript presented in an intelligible fashion and written in standard English?

Reviewer #1: No

Reviewer #2: Yes

5. Review Comments to the Author

Reviewer #1: The paper is interesting only one problem is the language. I suggest major revision of the English. The abstract, introduction, material and methods, discussion and conclusions are good structured and with interesting results.

Reviewer #2: Predicting surgically relevant outcomes is an important clinical problem to make better curation on the patient. Christensen et al compare several machine learning approaches that can predict patients’ surgically relevant outcomes including atrial fibrillation, venous thromboembolism, and pneumonia. They compared the performance by using different combinatorial features (SNPs/clinical information/both) and models (linear model, deep neural networks). Based on the prediction result, the author claimed that the deep neural network out-performs linear model. However, in my experience, I have many concerns.

Major concerns:

1. The author does not provide the script and code of the work. I suggest the author provide the code and the dependency for reproducibility. I also suggest author provide the preprocessed data for training model.

2. The feature selection process is not correct. In the machine learning process, the dataset should be split into a training set and a test set at the beginning. Based on the training set, the author can perform feature selection (SNP selection). The test set should be set aside until final testing to get the performance of the model. Otherwise, the information in the test set will be leaked to the model, which make the performance of the model overestimated. However, the author select the top 100 SNPs based on the whole dataset including training and test set. I think this might be the reason for the great AUC performance of the linear model. The performance of DNN seems not as good as the linear model.

3. The performance of the model should be measured by AUC rather than a recall value that depends on a specific cutoff. To measure the imbalance data, I suggest authors use the area under the precision-recall curve (PRAUC) instead, which is more sensitive to imbalance data. AUC and PRAUC can measure the global performance of machine learning better.

Minor concerns:

1. In the method session, the top feature selection process of 100 SNPs is merged into the session of the linear model, which makes the reader confused about whether the DNN uses all features or these selected features.

2. The author only provides the top feature of the model. I suggest the authors provide the information of positive or negative effects (coefficient and SHAP value) to the prediction value. Ensuring the supporting literature matches the effect is necessary.

6. PLOS authors have the option to publish the peer review history of their article (what does this mean?). If published, this will include your full peer review and any attached files.

Reviewer #1: No

Reviewer #2: No

---

## [Author Response · Author response to Decision Letter 0]

13 May 2024

Editorial comments:

Answer:

Thank you. We have updated the manuscript, figures and tables accordingly. 

Answer:

We have now made all queries and code available at https://github.com/satihest/surgical_complications. 

3. In the online submission form, you indicated that data is not publicly available but can be applied for at https://www.ukbiobank.ac.uk/enable-your-research/apply-for-access. Analytic methods will be made public at github.com at request. Requests to access these datasets should be directed to https://www.ukbiobank.ac.uk/enable-your-research/apply-for-access.

Answer:

Thank you. We have included a statement below:

The de-identified dataset used for this study can be obtained from the authors, provided written authorization from UK Biobank can be obtained. Authors are not allowed to share data without express permission from this governing body. The UK Biobank can be contacted at https://www.ukbiobank.ac.uk/learn-more-about-uk-biobank/contact-us. Enquiries from researchers about applying for access should be directed to the Access Management Team – email: access@ukbiobank.ac.uk. 

4. Please amend your manuscript to include your abstract after the title page.

Answer:

We have now invluded the abstract after the title page. 

5. We notice that your supplementary [figures/tables] are included in the manuscript file. Please remove them and upload them with the file type 'Supporting Information'. Please ensure that each Supporting Information file has a legend listed in the manuscript after the references list.

Additional Editor Comments :

In addition to address the critical comments from both Reviewers in full, I would encourage that the authors follow the latest version of the Polygenic Risk Score Reporting Standards (PMID: 33692554) to ensure the methodological transparency and scientific rigor in the revised manuscript.

Answer:

We have now deleted the supplementary material from the manuscript and included only titles/legends after references. Further, we have updated the manuscript accordingly with the Polygenic Score Reporting Standards. 

 

Reviewer 1:

R-1-1:

The paper is interesting only one problem is the language. I suggest major revision of the English. The abstract, introduction, material and methods, discussion and conclusions are good structured and with interesting results.

Answer:

We would like to thank the reviewer for taking the time to review our paper. We are happy that the reviewer found our paper interesting, and we appreciate the relevant feedback on the language. Accordingly, we have had the paper reviewed by an American physician scientist (Dr. Jaimie Chang, MD, resident physician at Rush Medical Center) and have updated the grammar and language accordingly to her suggestions. Please see the entire revised manuscript for the updates. 

Reviewer 2: 

Predicting surgically relevant outcomes is an important clinical problem to make better curation on the patient. Christensen et al compare several machine learning approaches that can predict patients’ surgically relevant outcomes including atrial fibrillation, venous thromboembolism, and pneumonia. They compared the performance by using different combinatorial features (SNPs/clinical information/both) and models (linear model, deep neural networks). Based on the prediction result, the author claimed that the deep neural network out-performs linear model. However, in my experience, I have many concerns.

R-2-1:

The author does not provide the script and code of the work. I suggest the author provide the code and the dependency for reproducibility. I also suggest author provide the preprocessed data for training model.

Answer:

First of all, we would like to thank the reviewer for taking the time to review our paper and for the relevant critique. We believe the suggestions have improved the manuscript. We agree that the code should be publicly available, and have therefore uploaded all Python code, shell scripts and UK Biobank queries used on the following Github account: https://github.com/satihest/surgical_complications

Unfortunately, UK Biobank data is not publicly available, but can be applied for by anyone. We cannot, without their permission, upload the unprocessed training data. 

R-2-2:

The feature selection process is not correct. In the machine learning process, the dataset should be split into a training set and a test set at the beginning. Based on the training set, the author can perform feature selection (SNP selection). The test set should be set aside until final testing to get the performance of the model. Otherwise, the information in the test set will be leaked to the model, which make the performance of the model overestimated. However, the author select the top 100 SNPs based on the whole dataset including training and test set. I think this might be the reason for the great AUC performance of the linear model. The performance of DNN seems not as good as the linear model.

Answer:

We agree, and we have updated the analyses accordingly. Please see the revised manuscript for the updated results. Overall, there were no major changes in the SNPs generated from doing the GWAS models on only the training data, and the models were therefore comparable in predictability. However, there were changes in the SNPs most important for the models. 

Please see below and in the revised results and discussion in the revised manuscript.

Changes to the manuscript, Page (P) 8, line (l) 19:

The cohorts were split into training/validation and test sets before, and only the training data was used for the initial GWAS-models. Relevant GWAS plots, including Manhattan and Quantile-Quantile (QQ) plots were generated using qqman

Changes to the manuscript, P 10, l 6:

Baseline characteristics are listed in table 1. The SNP model reached a ROC-AUC of 60.9% [95% CI, 60.6%-61.0%] 60.3 [95% CI, 60.0%-60.4%]. All individuals were classified as not having AF. The PRAUC was 0.09. The clinical model reached a ROC-AUC of 78.7% [95% CI, 78.7%-78.7%] with a recall of 9% and a precision of 53%. The PRAUC was 0.25. The combined model reached a ROC-AUC of 80.1% [95% CI, 80.0%-80.1%] with a recall of 9% and a precision of 57% 80.0% [95% CI, 79.9%-80.0%] with a recall of 9% and a precision of 56%. The PRAUC was 0.28.

Changes to the manuscript, P 12, l 6:

Baseline characteristics for VTE are listed in table 2. The SNP model reached a ROC-AUC of 59.6% [95% CI, 59.0%-59.7%] 60.1% [95% CI, 59.6%-60.4%]. All individuals were classified as not having VTE. The PRAUC was 0.04. The clinical model reached a ROC-AUC of 63.4% [95% CI, 63.2%-63.4%]. All individuals were classified as not having VTE. The PRAUC was 0.04. The combined model reached a ROC-AUC of 66.1% [95% CI, 65.7%-66.1%] 66.6% [95% CI, ]. All individuals were classified as not having VTE. The PRAUC was 0.05

Changes to the manuscript, P 14, l 5:

Baseline characteristics are listed in table 3. The SNP model reached a ROC-AUC of 57.3% [95% CI, 56.5%-57.4%] 50.1% [95% CI, 49.6%-50.6%]. All individuals were classified as not having pneumonia. The PRAUC was 0.04. The clinical model reached a ROC-AUC of 69.2% [95% CI, 69.1%-69.2%]. All individuals were classified as not having pneumonia. The PRAUC was 0.12. The combined model reached a ROC-AUC of 70.5% [95% CI, 70.2%-70.6%] with a recall of 0.01% and a precision of 0.4% 68.4% [95% CI, 68.0%-68.5%] with a recall of 0.01 and a precision of 0.5. The PRAUC was 0.11.

Changes to the manuscript, P 16, l 7:

The SNP model reached a ROC-AUC of 59.9% [95% CI, 58.6%-61.3%] 59.4% [95% CI, 58.2%-60.9%]in the test set. Recall was 36.9% 42.8% and precision was 9.3% 8.7%, with the area under the precision-recall curve (PRAUC) being 0.09. The clinical model reached a ROC-AUC of 78.8% [95% CI, 77.8%-79.8%] with recall and precision of 72.0% and 13.5%, respectively, with the PRAUC being 0.25. The combined model reached a ROC-AUC of 79.4% [95% CI, 78.8%-80.5%] 79.8 [95% CI, 78.8%-80.9%] with a recall and precision of 74.8% and 13.5%, respectively, with the PRAUC being 0.27.

Changes to the manuscript, P 18, l 4:

The SNP model reached a ROC-AUC of 60.0% [95% CI, 57.8%-61.8%] 51.5% [95% CI, 49.4%-53.4%] with a recall of 50.8% and precision of 4% and a PRAUC of 0.03. The clinical model reached a ROC-AUC of 63.2% [95% CI, 61.2%-65.0%] with a recall and precision of 67.5% and 4.0%, respectively, and a PRAUC of 0.05. The combined model reached a ROC-AUC of 65.4% [95% CI, 63.6%-67.2%] 62.6% [95% CI, 60.7%-64.5%] with a recall and precision 68.8% and 4.0%, respectively, and a PRROC of 0.05

Changes to the manuscript, P 19, l 5:

The SNP model reached a ROC-AUC of 55.5% [95% CI, 54.1%-56.9%] 51.0% [95% CI, 49.7%-52.4%] with a recall of 55.0% and precision of 5% and a PRAUC of 0.05. The clinical model reached a ROC-AUC of 69.7% [95% CI, 68.5%-70.8%] with a recall and precision of 67.7% and 7.4%, respectively and a PRAUC of 0.13. The combined model reached a ROC-AUC of 69.9% [95% CI, 68.7%-71.0%] 69.7% [95% CI, 68.6%-70.8%] with a recall and precision of 70.1% and 7.3%, respectively, and a PRAUC of 0.12.

Changes to the manuscript, P 21, l 6:

Overall, we found that adding SNP data to clinical risk prediction models enhanced the predictive power, and that the GLN approach seemed superior to a legacy linear risk prediction approach.

Changes to the manuscript, P 21, l 15:

The GLN-models performed better on recall and precision, and it was were able to classify positives correctly with just genomic information, however, the PRAUC was generally comparable between the deep learning and linear models with identical data. Given that the linear and GLN models utilize distinct tuning parameters for hard predictions, a direct comparison of recall and precision may not be critically significant, although the deep learning models generally performed better. However, as hard predictions are necessary in a clinical setting, a discussion is still warranted. Precision-recall curves are depicted in supplementary materials. 

Changes to the manuscript, P 21, l 22:

It is exemplified for AF, where the genetic linear model had a ROC-AUC of 60.9% [95% CI 69.6%-61.0%] while the GLN had a ROC-AUC of 59.9% [95% CI 5.8.6%-61.3%] 60.3% [95% CI, 60.0%-60.4%] while the GLN had a ROC-AUC of 59.4% [95% CI, 58.2%-60.9%]. The recall, however, was 0% and 36.9% 42.8%,

Changes to the manuscript, P 22, l 2:

On the contrary, the GLN-model had a recall of 36.9% 42.8% and therefore identifies around more than one third of cases correctly, which heightens the likelihood of clinical meaningful utilization considering only SNPs were included in the model. However, the precision was calculated low at 4%, 8.7%, which would lead to overdiagnosis and possibly overtreatment.

Changes to the manuscript, P 22, l 9:

It, however, cannot be ruled out that the differences between the models are not due to an inherent predictive advantage in the GLN-model, but simply due to different hyperparameter tuning, as indicated by the similar performance in PRAUC. 

Changes to the manuscript, P 22, l 21:

administrative codes are used for PC curation.(19)

GLN based models did, however, outperform linear approaches in terms of recall performance, might indicate that the ability to capture the effects of non-linear genetic traits on the overall phenotype, may be possible through this modelling approach. 

Changes to the manuscript, P 23, l 4:

Identified Single Nucleotide Polymorphisms

In the GLN-model, rs3807989 was the most activated SNP in regards of classifying individuals with AF. It is an intron variant in CAV1 which codes for a main component in caveolae plasma membrane and further acts as a tumor suppressor.(20, 21) It has been associated with a large variety of diseases including AF in numerous populations.(22, 23) Interestingly, the prevalence of the reference and risk allele is roughly equal which suggests the possibility of a relatively new mutation or that the risk variant has a different functional advantage which balances the selection. In the GLN model, rs17042171 was the most activated SNP in regards of classifying individuals with AF. It is an intergenic variant downstream from PITX2 which codes for the paired-like homeodomain transcription factor 2.(24, 25) The risk allele is moderately prevalent in the European population with a prevalence of 15%. suggesting a negative selection pressure of the risk allele (26)

Changes to the manuscript, P 23, l 16:

a variant near 4q25, which has been extensively associated with AF in a variety of populations and is also in close proximity to PITX2.(25, 27), The variants most highly associated with AF in the linear model was also rs17042171, also a variant near 4q25. The alternative allele has worldwide prevalence of up to 16% and 13% in the European population, which makes the risk variant very common, although not equal to the reference allele suggesting a negative selection pressure of the risk allele.(26)

Changes to the manuscript, P 23, l 25:

Besides the top variant, rs17042171, Tthe difference in which variants show importance for the GLN and linear model, respectively, and that the GLN-models in general performed significantly better in recall compared with the matching linear, shows that non-linear interactions between genes which are potentially of great importance in the risk of a particular trait

Changes to the manuscript, P 24, l 8:

the genes have previously been described to be in a direct pathway or in any kind of interaction. Interestingly, Gao et al. showed that the level of caveolin-1 determines the level of product of KCNN1 which previously has been highly associated with AF in several GWAS-studies.(22, 28-30) 

Changes to the manuscript, P 24, l 11:

The SNPs with highest importance for classifying VTE in the GLN-model was rs505922 rs8176745, a synonymous variant in, an intron variant in ABO.(31) The variant has previously been associated variant has, to our knowledge, not previously been associated with VTE.(6)

Changes to the manuscript, P 24, l 19:

The SNP that had the highest feature importance in the GLN-model for classifying bacterial pneumonia was rs11080143 rs17851582 for the SNP-model and rs9353801 for the combined model

Changes to the manuscript, P 24, l 22:

but rather due to chance alone. rs17851582 is a missense variant in GAMT which codes for the liver enzyme guanidinoacetate methyltransferase. Deficiency of the enzyme can affect brain and muscle development and lead to severe neurological problems including epilepsy.(32) However, the specific variant in our analysis is deemed to be benign.(33) rs9353801 was most important in the combined model. It h

---

## [Decision Letter · Decision Letter 1]

5 Jun 2024

PONE-D-23-35139R1An assessment of the value of deep neural networks in genetic risk prediction for surgically relevant outcomesPLOS ONE

Dear Dr. Sillesen,

Thank you for submitting your manuscript to PLOS ONE. After careful consideration, we feel that it has merit but does not fully meet PLOS ONE’s publication criteria as it currently stands. Therefore, we invite you to submit a revised version of the manuscript that addresses the points raised during the review process.

**As pointed by one reviewer, the code for this paper is currently not available to the public: https://github.com/satihest/surgical_complications.**

**Before I can reach a final decision, I would value the opportunity to review the code. This will allow me to verify its alignment with the conclusions of the paper and the policies of the journal.**

We look forward to receiving your revised manuscript.

Kind regards,

Xiang Zhu

Academic Editor

PLOS ONE

Journal Requirements:

Additional Editor Comments:

As pointed by one reviewer, the code for this paper is currently not available to the public: https://github.com/satihest/surgical_complications.

Before I can reach a final decision, I would value the opportunity to review the code. This will allow me to verify its alignment with the conclusions of the paper and the policies of the journal.

Reviewers' comments:

Reviewer's Responses to Questions

**Comments to the Author**

1. If the authors have adequately addressed your comments raised in a previous round of review and you feel that this manuscript is now acceptable for publication, you may indicate that here to bypass the “Comments to the Author” section, enter your conflict of interest statement in the “Confidential to Editor” section, and submit your "Accept" recommendation.

Reviewer #1: All comments have been addressed

Reviewer #2: All comments have been addressed

2. Is the manuscript technically sound, and do the data support the conclusions?

Reviewer #1: Yes

Reviewer #2: Yes

3. Has the statistical analysis been performed appropriately and rigorously? 

Reviewer #1: Yes

Reviewer #2: Yes

4. Have the authors made all data underlying the findings in their manuscript fully available?

Reviewer #1: Yes

Reviewer #2: No

5. Is the manuscript presented in an intelligible fashion and written in standard English?

Reviewer #1: Yes

Reviewer #2: Yes

6. Review Comments to the Author

**Reviewer #1: **The paper now is very good in all parts. In particular the discussion has a significant value for the scientific community.

**Reviewer #2:** The issues have been improved. But I cannot access the code: https://github.com/satihest/surgical_complications

Please check if it is public.

7. PLOS authors have the option to publish the peer review history of their article (what does this mean?). If published, this will include your full peer review and any attached files.

Reviewer #1: No

Reviewer #2: No

---

## [Author Response · Author response to Decision Letter 1]

12 Jun 2024

Dear Editors

We apologize for the issue with the broken Github link. It turns out that the github repository was set to private rather than public access. This has now been ammeneded, and the project code is now freely available via the link https://github.com/satihest/surgical_complications. 

Again, we apologize for this error and the delays that it has caused. 

Please feel free to reach out to us if further information is needed. 

Kind Regards

Martin Sillesen

---

## [Decision Letter · Decision Letter 2]

2 Jul 2024

An assessment of the value of deep neural networks in genetic risk prediction for surgically relevant outcomes

PONE-D-23-35139R2

Dear Dr. Sillesen,

We’re pleased to inform you that your manuscript has been judged scientifically suitable for publication and will be formally accepted for publication once it meets all outstanding technical requirements.

Kind regards,

Xiang Zhu

Academic Editor

PLOS ONE

Additional Editor Comments (optional):

Reviewers' comments:

Reviewer's Responses to Questions

**Comments to the Author**

1. If the authors have adequately addressed your comments raised in a previous round of review and you feel that this manuscript is now acceptable for publication, you may indicate that here to bypass the “Comments to the Author” section, enter your conflict of interest statement in the “Confidential to Editor” section, and submit your "Accept" recommendation.

Reviewer #2: All comments have been addressed

2. Is the manuscript technically sound, and do the data support the conclusions?

Reviewer #2: Yes

3. Has the statistical analysis been performed appropriately and rigorously? 

Reviewer #2: Yes

4. Have the authors made all data underlying the findings in their manuscript fully available?

Reviewer #2: Yes

5. Is the manuscript presented in an intelligible fashion and written in standard English?

Reviewer #2: Yes

6. Review Comments to the Author

Reviewer #2: The github link is available now. Necessary information has been included in github. Issues have been improved.

7. PLOS authors have the option to publish the peer review history of their article (what does this mean?). If published, this will include your full peer review and any attached files.

Reviewer #2: No

---

## [Editor Report · Acceptance letter]

4 Jul 2024

PONE-D-23-35139R2 

PLOS ONE

Dear Dr. Sillesen, 

I'm pleased to inform you that your manuscript has been deemed suitable for publication in PLOS ONE. Congratulations! Your manuscript is now being handed over to our production team.

Kind regards, 

on behalf of

Dr. Xiang Zhu 

Academic Editor

PLOS ONE